# The Role of miR-155 in Modulating Gene Expression in CD4+ T Cells: Insights into Alternative Immune Pathways in Autoimmune Encephalomyelitis

**DOI:** 10.3390/ijms252111355

**Published:** 2024-10-22

**Authors:** Maria Cichalewska-Studzinska, Jacek Szymanski, Emilia Stec-Martyna, Ewelina Perdas, Miroslawa Studzinska, Hanna Jerczynska, Dominika Kulczycka-Wojdala, Robert Stawski, Marcin P. Mycko

**Affiliations:** 1Research Laboratory CoreLab, Medical University of Lodz, 92-215 Lodz, Poland; jacek.szymanski@umed.lodz.pl (J.S.); emilia.stec-martyna@umed.lodz.pl (E.S.-M.); miroslawa.studzinska@umed.lodz.pl (M.S.); hanna.jerczynska@umed.lodz.pl (H.J.); dominika.kulczycka-wojdala@umed.lodz.pl (D.K.-W.); 2Department of Biostatistics and Translational Medicine, Medical University of Lodz, 92-215 Lodz, Poland; ewelina.perdas@umed.lodz.pl; 3Department of Clinical Physiology, Medical University of Lodz, 92-215 Lodz, Poland; robert.stawski@umed.lodz.pl; 4Department of Neurology, Laboratory of Neuroimmunology, University of Warmia and Mazury in Olsztyn, 10-082 Olsztyn, Poland; marcin.mycko@uwm.edu.pl

**Keywords:** autoimmune disorders, CD4+ T cells, experimental autoimmune encephalomyelitis (EAE), immunization, KO mice, miR-155, MOG

## Abstract

CD4+ T cells are considered the main orchestrators of autoimmune diseases. Their disruptive effect on CD4+ T cell differentiation and the imbalance between T helper cell populations can be most accurately determined using experimental autoimmune encephalomyelitis (EAE) as an animal model of multiple sclerosis (MS). One epigenetic factor known to promote autoimmune inflammation is miRNA-155 (miR-155), which is significantly upregulated in inflammatory T cells. The aim of the present study was to profile the transcriptome of immunized mice and determine their gene expression levels based on mRNA and miRNA sequencing. No statistically significant differences in miRNA profile were observed; however, substantial changes in gene expression between miRNA-155 knockout (KO) mice and WT were noted. In miR-155 KO mice, mRNA expression in CD4+ T cells changed in response to immunization with the myeloid antigen MOG_35-55_. After restimulation with MOG_35-55_, increased *Ffar1* (free fatty acid receptor 1) and *Scg2* (secretogranin-2) expression were noted in the CD4+ T cells of miR-155-deficient mice; this is an example of an alternative response to antigen stimulation.

## 1. Introduction

Myelin oligodendrocyte glycoprotein (MOG) is a crucial component of the myelin sheath within the central nervous system (CNS) and plays a significant role in autoimmune diseases such as multiple sclerosis (MS) [1]. MS is an inflammatory demyelinating disorder characterized by the immune system attacking the myelin sheath. This autoimmune pathology is driven largely by myelin-reactive CD4+ T helper (Th) cells, which are critical in the initiation and progression of the disease. Among the various subsets of Th cells, Th17 cells have been identified as key players in MS [2]. These cells are distinguished by the expression of several transcription factors, including retinoic acid receptor-related orphan receptor alpha (ROR-α), ROR-γt, signal transducer and activator of transcription 3 (STAT3), and basic leucine zipper ATF-like transcription factor (BATF). Additionally, Th17 cells produce the proinflammatory cytokine interleukin 17 (IL-17), which is instrumental in mediating inflammatory responses. The significance of Th17 cells in the pathogenesis of MS has been highlighted through research, particularly studies using experimental autoimmune encephalomyelitis (EAE), an animal model that closely mimics human disease. EAE studies have shown that mice with reduced or dysfunctional Th17 cells, especially those lacking key cytokines such as IL-6, IL-21, or IL-23, exhibit notable resistance to the development of EAE [3,4].

To better understand autoimmune mechanisms, it is essential to explore MOG-specific immune responses. MOG restimulation in CD4+ T cells has become a valuable model for studying autoimmune reactions and the underlying mechanisms of immune dysregulation. Reactivating MOG-specific T cells may indicate how these cells contribute to autoimmune pathology and identify potential intervention points for therapeutic strategies.

In recent years, microRNAs (miRNAs) have garnered attention for their role in regulating various biological processes, including immune responses and T cell differentiation [5]. These small, noncoding RNA molecules typically consisting of 19 to 24 nucleotides, are involved in post-transcriptional gene regulation. Several miRNAs have been implicated in autoimmune diseases, particularly in relation to Th17 cells and autoimmune demyelination. Among these, miR-155 has emerged as a key regulator of inflammatory responses and Th17 cell differentiation [6]. Research has indicated that miR-155 plays a crucial role in the development and function of Th17 cells by targeting specific genes and regulating cytokine production. The absence or inhibition of miR-155 has been shown to delay the onset and reduce the severity of EAE, suggesting its potential as a therapeutic target [7]. In particular, miR-155 facilitates Th17 cell development by modulating the expression of related genes and is necessary for the secretion of Th17-related cytokines in dendritic cells (DCs) [8].

Our study aims to investigate the alternative responses of CD4+ T cells to MOG restimulation within the context of miR-155 deficiency in miR-155 knockout mice (KO). By examining how the lack of miR-155 influences the behavior of CD4+ T cells, it aims to elucidate the role of this miRNA in regulating autoimmune responses. This research is expected to provide valuable insights into the mechanisms by which miR-155 modulates immune function and its potential implications for the development of novel therapeutic approaches for autoimmune diseases including MS. Understanding how miR-155 affects the immune response to MOG can help in identifying new targets for intervention and contribute to advancing our knowledge of autoimmune disease pathogenesis.

## 2. Results

### 2.1. Study Design (Workflow)

In order to induce immunological response, mice were first transfected with myelin antigen MOG_35-55_ in complete Freund’s adjuvant (CFA). Following this, cells extracted from lymph nodes were cultured in vitro for 72 h either with MOG_35-55_ stimulation or without any stimulation. In the next step, CD4 T cells were sorted, and RNA was isolated from these cells to prepare for next generation sequencing (NGS) (Figure 1).

### 2.2. Mutagenesis and miR-155 KO Mice Generation

To determine which genes are regulated by miR-155 in CD4+ T cells in response to myelin antigen stimulation, miR-155-deficient mice were generated using the CRISPR/Cas9 method. Two guide RNAs (gRNAs) were used to find complementary sequences in the genome where Cas9 protein binds to DNA strands close to the protospacer adjacent motif (PAM) and induces a double-strand break in the template DNA (Figure 2a). gRNAs applied to guide Cas9 to delete the miR-155 sequence: (1) GTTGCATATCCCTTATCCTC and (2) GACTTGTCATCCTCCCACGG. The deleted region spanned approximately 145 base pairs, including 34 bases upstream of the miR-155 coding sequence and 46 bases downstream (Figure 2a). To confirm the successful deletion of the targeted genomic sequence, PCR was performed. The PCR product was not sequenced, but the size of the product was confirmed on electrophoresis agarose gel; the results confirmed the deletion of miR-155, thus allowing the identification of wild-type (WT) (+/+), heterozygous (+/−), and null (−/−) miR-155 mice. Subsequently, the copy numbers of miRNA-155-3p and miRNA-155-5p were quantified in WT, homozygous, and miR-155 KO mice (Figure 2b). RNA isolated from splenocytes was subjected to reverse transcription with specific primers and PCR using Taqman probes for miR-155-3p and miR-155-5p. No copies of miR-155-3p were detected in samples from KO mice, while 3.6 copies were found in KO mice compared to 675 copies in WT mice (Figure 2c).

### 2.3. MOG Restimulation Induces Changes in Gene Expression Profiles in T Cells from miRNA-155-Sufficient and miRNA-155-Deficient Mice

To identify changes in the miRNA profile of miR-155-deficient mice in response to myelin autoantigen, WT and miR-155 KO mice were immunized with MOG_35-55_. Thirteen days later, the lymph node cells were stimulated in vitro with MOG_35-55_ or left unstimulated in culture media for 72 h. Next, the, CD4+ T cells were purified using magnetic columns and used as a source of RNA for NGS sequencing.

The miRNA profiles of MOG restimulated CD4+ T cells from miR-155 KO mice were compared with CD4+ T cells from WT mice without in vitro stimulation using small RNA (sRNA) sequencing (Appendix A). Generally, no significant differences in the miRNA profile were noted between the miR-155-deficient and miR-155-sufficient mice; however, miRNA-155 copies were observed in the KO mice. These results confirm the efficacy of miR-155 coding sequence excision (Figure 3a).

Any changes in gene expression occurring in CD4+ T cells after MOG_35-55_ restimulation were identified by mRNA NGS sequencing (Appendix A). The results indicate that the samples demonstrated diverse molecular interactions in response to antigen stimulation (Figure 3b). Since miRNAs typically decrease the expression of their target genes, the analysis focused on genes that were upregulated in the absence of miR-155 (Figure 3c).

### 2.4. miRNA-155 Has Multiple Gene Targets

Any potential targets of miRNA-155 among the tested genes were identified using the bioinformatics tool miRNet 2.0 (http://www.mirnet.ca/, accessed on 12 June 2024), which generated a network of regulatory interactions (Figure 4). Based on the miRNet data and a literature review, the identified genes were selected for further analysis.

### 2.5. Ffar1 and Scg2 Upregulation in miR-155-Deficient CD4+ T Cells

Data from mRNA and miRNA sequencing of changes between KO MOG and WT are attached in the Appendix A, including differential expression analysis between conditions or groups with *p* adjusted < 0.05. In CD4+ T cells after MOG restimulation from miR-155-deficient mice to control WT, we observed upregulation of 23 genes and 16 genes were downregulated.

To identify the regulators of CD4+ T cells in response to MOG restimulation, NGS sequencing results were set to be confirmed with qPCR. Among the tested candidates, two genes, *Ffar1* and *Scg2*, showed increased expression (Figure 5).

The expression of the *Ffar1* and *Scg2* genes was increased by approximately 6-fold and 9-fold, respectively, in miR-155-deficient CD4+ T cells stimulated with MOG compared to WT cells.

Among the miR-155-deficient CD4+ T cells, the genes found to be upregulated in response to myelin antigen stimulation may play regulatory roles and be involved in various biological processes. However, a better understanding of these processes requires further study (Figure 6).

## 3. Discussion

The understanding of the etiology of autoimmune demyelination in MS remains incomplete. However, it is widely accepted that a central role in disease initiation is played by an autoimmune response targeting epitopes on the myelin sheath within the CNS. Indeed, a complex milieu formed by immune system dysregulation, genetic susceptibility, and environmental factors, including infections and vitamin D levels, is believed to contribute to the complex pathophysiology of MS. In the experimental autoimmune EAE model, a well-established murine model of MS, myelin-specific T lymphocytes are activated in the periphery and migrate into the CNS, crossing the blood–brain barrier—a process facilitated by pertussis toxin administration [9].

An area of particular interest in the study of CNS demyelination is that of the role played by CD4+ T cells. Although their significance has been recognized, their specific contributions have yet to be thoroughly explored. Recent studies have examined the response of CD4+ T cells to the MOG_35-55_ peptide, demonstrating abundant interferon γ (IFN-γ)-producing CD4+ T cells in the spleens of MOG_35-55_-immunized mice [10]. Additionally, MOG_35-55_ stimulation leads to increased expression of pro-inflammatory cytokines such as IL-17, tumor necrosis factor α (TNFα), and IL-22, indicating the involvement of these cells in inflammatory processes [11].

miR-155 modulates the IL-17/IL-23 axis in autoimmune diseases [12], and is believed to influence the M1/M2 balance in macrophages by modulating the IL-13 effect [13]. In such cases, the degeneration of the myelin sheaths is believed to result from an autoimmune attack on a myelin autoantigen mediated by CD4+ T-cells, especially Th17 cells producing IL-17 [14]. CD4+ T cells transfected with mimics for miR-155 showed upregulation in IFN-γ and IL-17 levels, while IFN-γ triggers microglia apoptosis as the activation-induced cell death [14,15].

Moreover, miR-155 modulates the level of IL-23 in CNS-resident microglia, which could play a critical role in the pathogenesis of MS [12,16].

It influences various processes, including cell proliferation, cytokine production, and the differentiation of naive CD4+ T cells into Th1 and Th17 subsets [17], which are known to play key roles in autoimmune diseases like MS. Beyond CD4+ T cells, miR-155 also impacts other immune cells, such as B cells and DCs, contributing to the pathogenesis of autoimmune diseases by promoting aberrant immune responses [18].

Rodriguez et al. demonstrated that miR-155-deficient mice exhibit altered immune responses, particularly in lymphocytes and DCs [19]. Notably, miR-155-deficient mice show significant resistance to EAE, characterized by a reduced severity and delayed onset compared to WT controls [7,8]. Moreover, anti-miR-155 treatment has been shown to attenuate the clinical course of EAE, suggesting that it may have potential as a therapeutic target [7]. However, the exact mechanism of action of miR-155 remains unclear.

Although numerous studies have examined the changes induced by activated CD4+ T cells within the central nervous system [20,21], fewer have focused on changes occurring in the periphery; these are particularly significant because oligodendrocyte glycoprotein (MOG)-specific tolerance can be established by ectopic expression of MOG in the immune organs [22]. Our present study focuses on CD4+ T cells in the peripheral lymph nodes, rather than those that cross the blood–brain barrier. In the present study, a novel miR-155 KO mouse was chosen as a research model to better understand the role of this miRNA as a critical regulator in the response of peripheral CD4+ T cells to myelin antigen stimulation. Successful deletion of miR-155 was confirmed by both modification of the genomic locus and by demonstration of the absence of miR-155 transcripts. Moreover, sequencing of the PCR product would further confirm successful deletion of the selected region. It was found that intraperitoneal administration of myelin antigen in miR-155-deficient mice correlates with significant changes in gene expression; more specifically, that *Ffar1* and *Scg2* are upregulated in peripheral CD4+ T cells following MOG_35-55_ stimulation. *Ffar1*, a G-protein-coupled receptor, is primarily known for its role in glucose homeostasis through insulin secretion. However, it is also expressed in the CNS, suggesting a potential link between metabolic and immune responses in autoimmune conditions [23]. *Ffar1* activation is also involved in immunomodulation by influencing in the differentiation of monocytes and Kupffer cells into M2 macrophages n [24].

*Scg2*, part of the chromogranin-secretogranin family [25], has been implicated in autoimmunity, particularly in the recognition of autoantigens by CD8+ T cells; this suggests it may have a potential role in autoimmune reactions against pancreatic beta cells, as indicated by research into peptides derived from *Scg2* [26]. Moreover, increased levels of *Scg2* have been associated with favorable clinical outcomes in certain cancers, such as kidney renal clear cell carcinoma, and in calcific aortic valve disease (CAVD) patients [27].

miR-155 modulates the IL-17/IL-23 axis in autoimmune diseases [12]. It is also involved in regulating the M1/M2 balance in macrophages by modulating the effect of IL-13 [13]. Damage to the myelin sheaths is a consequence of an autoimmune process directed against a putative myelin autoantigen mediated by CD4+ T-cells, especially Th17 cells producing IL-17 [14]. CD4+ T cells transfected with mimics for miR-155 showed upregulation in IFN-γ and IL-17 levels, while IFN-γ triggers microglia apoptosis as a form of activation-induced cell death [14,15].

Conversely, miR-155 modulates the level of IL-23 in CNS-resident microglia, which could be critical for the pathogenesis of MS [12,16].

Our findings suggest that miR-155 could modulate immune responses through an alternative cytokine response pathway involving *Ffar1* and *Scg2*. This adds a novel perspective to the current understanding of miR-155, which has traditionally been linked to cytokine regulation. The upregulation of these genes in miR-155-deficient mice may represent alternative mechanisms contributing to the modulation of autoimmune responses [28].

There are a few limitations to the study. Although the findings indicate *Ffar1* and *Scg2* to be upregulated in miR-155-deficient CD4+ T cells, further investigation is necessary to clarify the functional significance of these gene responses and their potential roles in immune regulation. In addition, the research primarily focuses on immune responses in peripheral lymphoid tissues, potentially overlooking crucial processes occurring in the CNS, where EAE has significant implications. This may hinder a comprehensive understanding of disease progression within the CNS. Additionally, the use of a murine model may not entirely reflect the pathophysiology of human MS, which could impact the broader applicability of the findings. Validation with a larger sample number is also recommended to enhance the robustness of the conclusions.

Future studies should aim to validate these findings across different models and further investigate the functional roles of *Ffar1* and *Scg2* in human CNS inflammation and autoimmunity. While translating these observations into clinical applications will present a challenge, such innovative approaches are demanded by the current lack of progress in new therapeutic strategies for MS.

## 4. Materials and Methods

### 4.1. Mice

The miR-155-deficient mouse line was developed by the Genome Engineering Facility at the International Institute of Molecular and Cell Biology in Warsaw. The line was created using clustered regularly interspaced short palindromic repeats (CRISPR)/CRISPR-associated protein 9 (Cas9) in a C57BL6/JRj genetic background. The mice were housed at the Animal Facilities of the Medical University of Lodz. A total of 47 mice were used in the study: 24 miR-155 KO mice and 23 miR-155-sufficient mice (one mouse died during the experiment). All mice were female and aged 8–12 weeks. All animal experiments were performed in compliance with the relevant laws and institutional guidelines and were approved by ethics committees. The animal study protocol was approved by the local ethics committee at the Medical University of Lodz (decisions 69/ŁB06/2015—7 December 2015 and 10/LB 261/2023—6 February 2023) for studies involving animals. Research conducted on GMO mice was also approved by the Polish Ministry of the Environment (DOP-GMO.431.226.2017 decision no 175/2017).

### 4.2. Genotyping

The absence of the miR-155 sequence in miR-155 KO mice was confirmed by genotyping using the primers F: CAGAGCTCTTTTCTTTCAAAGCTG and R: AGATGTTGTTTAGGATACTGCTG. The primers were designed using the SnapGene program and their specificity was verified using BLAST software v 2.12.0. DNA isolation was performed using the HotSHOT method [29] with minor modifications. Briefly, a tail biopsy was placed in a 1.5 mL Eppendorf tube and covered with 165 µL of lysis buffer (25 mM NaOH, 0.2 mM EDTA). The samples were incubated for 25 min at 95 °C, then for 10 min at 4 °C. Following this, samples were centrifuged for 10 s at 400× *g*, and 165 µL of neutralizing buffer (40 mM Tris-HCl, pH 5) was added. A 1 µL sample of the resulting DNA was added to 15 µL of amplification mix containing Dream Taq Polymerase (ThermoFisher Scientific, Waltham, MA, USA), Green Dream Taq Buffer, primers (10 µM each), dNTPs (10 mM, ThermoFisher Scientific, MA, USA), and water. PCR was performed on the Gene Amp PCR System 9700 (Applied Biosystems, ThermoFisher Scientific, Waltham, MA, USA) according to the manufacturer’s instructions with the following parameters: an initial step at 98 °C for 3 min and 15 s, followed by 35 cycles of 13 s at 98 °C, 17 s at 63 °C, 30 s at 72 °C, and a final extension at 72 °C for 6 min. PCR products were analyzed using 1.5% agarose gel electrophoresis in tris-acetate-EDTA (TAE) buffer.

### 4.3. Immunization and Lymph Nodes Cells Culture

The miR-155-sufficient mice and miR-155 KO mice, aged 8–12 weeks, were immunized subcutaneously on their abdominal flanks with 0.15 mg of MOG_35-55_ peptide (MEVGWYRSPFSRVVHLYRNGK, purity > 95%, AnaSpec, Fremont, CA, USA) in 150 µL of complete Freund’s adjuvant (CFA; Sigma-Aldrich, St. Louis, MO, USA) containing 0.75 mg of *Mycobacterium tuberculosis* (BD Difco, Waltham, MA, USA). After 13 days, axillary and inguinal lymph nodes were isolated, and single-cell suspensions were prepared. Cells were cultured in vitro at a density of 5 × 10^6^ cells per ml in 200 µL of IMDM (Iscove’s Modified Dulbecco’s Medium, Sigma-Aldrich) in U-bottomed 96-well microtiter plates. The cells were cultured for 72 h, either stimulated with MOG_35-55_ (20 µg/mL) or left unstimulated. Cell viability was assessed using propidium iodide (CytoFLEX, Beckman Coulter, Indianapolis, IN, USA). Subsequently, cells were sorted using indirect magnetic sorting with the CD4+ T Cells Isolation Kit II (Miltenyi Biotech, Bergisch Gladbach, Germany) according to the manufacturer’s instructions.

### 4.4. RNA Isolation and Preparation for Sequencing

Total RNA was extracted from cell cultures using the miRNeasy Mini Kit (Qiagen, Hilden, Germany) following the manufacturer’s instructions, including DNase treatment to eliminate genomic DNA. The final RNA was eluted in 30 μL of RNase-free water. RNA concentration was measured with a PicoDrop spectrophotometer (PicoDrop Limited, Hinxton, UK) and the Qubit RNA BR Assay (ThermoFisher Scientific, Waltham, MA, USA). RNA quality was assessed using a Bioanalyzer 2100 (Agilent Technologies, Waldbronn, Germany) with Pico RNA assays, and samples with RNA integrity numbers (RINs) above 8 were qualified for sequencing.

### 4.5. Absolute Gene Expression with Digital Quantitative PCR

Gene expression analysis of murine splenocytes was conducted using the QX200 Droplet Digital PCR (ddPCR) system (Bio-Rad, Hercules, CA, USA). TaqMan MicroRNA assays used for this research were mmu-miR-155-5p (Assay ID 002571) and mmu-miR-155-5p (Assay ID 64539_mat) (Life Technologies, ThermoFisher Scientific, Waltham, MA, USA). Reverse transcription was performed on 10 ng of total RNA in 15 µL reactions using TaqMan^®^ MicroRNA Reverse Transcription Kit (Applied Biosystems, ThermoFisher Scientific, Waltham, MA, USA) according to the manufacturer’s instructions. The ddPCR reaction mixture consisted of 10 µL of 2× ddPCR Supermix for Probes (no dUTP) (Cat. No. 1863024, Bio-Rad, Hercules, CA, USA), 1 µL of FAM-labeled fluorescent probe for miR-155-3p or miR-155-5p (ThermoFisher Scientific), 1.33 µL of cDNA, and 7.67 µL of DNase/RNase-free water. Droplet generation was carried out using the QX200 Droplet Generator (Bio-Rad, Hercules, CA, USA).

In this process, 20 µL of the reaction mixture and 70 µL of droplet generation oil for probes (Bio-Rad, Hercules, CA, USA) were combined in DG8 cartridges (Bio-Rad, Hercules, CA, USA). The resulting mixture was then transferred to a 96-well plate (Bio-Rad, Hercules, CA, USA), heat-sealed with aluminum foil using a PX1 PCR Plate Sealer (Bio-Rad, Hercules, CA, USA), and subjected to PCR in a T100 Thermal Cycler (Bio-Rad, Hercules, CA, USA). Fluorescent signals were detected with the QX200 Droplet Reader (Bio-Rad, Hercules, CA, USA), and positive droplets were distinguished from negative ones based on a threshold set at 4000 fluorescence (FAM) Amplitude (Figure 2c, top chart).

### 4.6. mRNA Sequencing

Quality control: Eight RNA samples (two from each: WT, WT_MOG, KO, and KO_MOG) were rigorously assessed according to standards detailed by Novogene. All samples were analyzed by electrophoresis on a 1% agarose gel at 180V for 16 min and by RNA Integrity test on a Bioanalyzer 2100 (Agilent Technologies, Waldbronn, Germany); the results indicate that all were suitable for library construction and sequencing. Each RNA sample was derived from sorted CD4+ T cells of three or four mice.

#### 4.6.1. Library Construction, Quality Control, and Sequencing

Library preparation was performed using the Novogene NGS RNA Library Prep Set (PT042). Genes identified as differentially expressed had an adjusted *p*-value < 0.05, as determined by DESeq2. Briefly, messenger RNA was purified from total RNA using poly-T oligo-attached magnetic beads. After fragmentation, first-strand cDNA was synthesized using random hexamer primers, followed by the second-strand cDNA synthesis [30]. The library was ready after end repair, A-tailing, adapter ligation, size selection, amplification, and purification. The library was assessed for quantity using Qubit and real-time PCR and for size distribution using a Bioanalyzer 2100. Quantified libraries were pooled and sequenced on the Illumina NovaSeq 6000 Sequencing System with paired-end reads of 150 nucleotides, achieving an average of 20 million read pairs per sample, based on effective library concentration and data requirements.

#### 4.6.2. Bioinformatics Analysis

Raw reads in FASTQ format were initially processed using custom Perl scripts. Briefly, the data were cleaned by removing reads with adapters, reads containing poly-N sequences, and low-quality reads. All subsequent analyses were conducted on these high-quality clean reads. Reference genome (ensembl_mus_musculus_grcm38_p6_gca_000001635_8) and gene model annotation files were directly downloaded from the ensemble. The reference genome index was created using Hisat2 v2.0.5, and paired-end clean reads were aligned to this reference genome with the same software [31]. Transcript abundance was quantified with FeatureCounts v1.5.0-p3 [32]. FPKM (fragments per kilobase of transcript per million mapped reads) for each gene was calculated based on gene length and read count mapped to this gene. Differentially expressed genes (DEGs) were analyzed using the DESeq2 package, with gene expression normalized using the relative-log-expression (RLE) method [33,34]. *p*-Values were adjusted for false discovery rate using the Benjamini and Hochberg approach [35]. Gene ontology (GO) [36] enrichment analysis, accounting for gene length bias, and KEGG [37] pathway analysis of DEGs were performed using the clusterProfiler R package v.4.9.1. GO terms with adjusted *p*-values < 0.05 were considered significantly enriched. The GO and KEGG datasets were also used independently for gene set enrichment analysis (GSEA).

### 4.7. miRNA Sequencing

#### 4.7.1. Quality Control

For miRNA sequencing, six RNA samples were isolated from 24 KO mice, and another six samples from 23 WT mice. Therefore, a total of 12 RNA samples (three from each group: WT, WT_MOG, KO, and KO_MOG) were isolated. Each RNA sample was isolated from lymphocytes derived from three or four mice, respectively. RNA samples were thoroughly evaluated according to Novogene’s quality standards. All samples met the requirement of library construction and sequencing after electrophoretic analysis on 1% agarose gel at 180 V for 16 min and an RNA integrity test performed on a Bioanalyzer 2100. Each RNA sample was derived from sorted CD4+ T cells from three or four mice.

#### 4.7.2. Library Construction, Quality Control and Sequencing

Libraries were prepared using the NEB Next^®^ Multiplex Small RNA Library Prep Set for Illumina^®^ (Set 1) (Cat No. E7300). RNA library preparation and sequencing were performed by Novogene Co., Ltd. (Beijing, China). Significant differences in miRNA expression were confirmed based on a corrected *p*-value of 0.05. Briefly, 3′ and 5′ adaptors were ligated to the respective ends of small RNA molecules. First-strand cDNA was synthesized after hybridization with a reverse transcription primer. The double-stranded cDNA library was then generated through PCR enrichment. Following purification and size selection, libraries with insert sizes of 18–40 bp were prepared for sequencing on an Illumina platform with SE50. Expression levels of known and unique miRNAs in each sample were statistically analyzed and normalized using transcripts per million (TPM). The library was quantified using Qubit and real-time PCR and assessed for size distribution with a Bioanalyzer 2100.

#### 4.7.3. Bioinformatics Analysis for miRNA

Raw FASTQ data (raw reads) were initially processed using custom Perl and Python scripts. In this step, clean data (clean reads) were obtained by removing reads with poly-N sequences, 5′ adapter contaminants, without 3′ adapters or insert tags, containing poly A, T, G, or C sequences, and low-quality reads from raw data. Small RNA tags were then mapped to a reference sequence using Bowtie, without allowing mismatches, to analyze their expression and distribution on the reference [38]. Mapped small RNA tags were used to identify known miRNAs. miRBase20.0 was used as reference, modified software mirdeep2 [39] and srna-tools-cli were used to obtain the potential miRNA and draw the secondary structures. miRNA expression levels were estimated using TPM. Differential expression analysis between conditions or groups was conducted with the DESeq R package (1.8.3), with *p*-values adjusted using the Benjamini and Hochberg method. Gene ontology (GO) enrichment analysis was performed on target gene candidates of differentially expressed miRNAs using GOseq with the Wallenius non-central hypergeometric distribution [36]. KEGG [40] is a database resource for understanding high-level functions and utilities of the biological system, such as the cell, the organism and the ecosystem, from molecular-level information, especially large-scale molecular datasets generated by genome sequencing and other high-throughput experimental technologies (http://www.genome.jp/kegg/, accessed on 15 June 2023). KOBAS [41] software (http://genome.cbi.pku.edu.cn/download.html) was used to test the statistical enrichment of the target gene candidates in KEGG pathways.

### 4.8. Reverse Transcription and Real-Time Quantitative PCR (RT-qPCR)

Reverse transcription was performed on the Gene Amp PCR System 9700 using the High Capacity cDNA Reverse Transcription Kit with RNase Inhibitor (Applied Biosystems, ThermoFisher Scientific, Waltham, MA, USA) according to the manufacturer’s instructions. The obtained cDNA samples were stored at −80 °C.

Gene expression analysis was conducted with TaqMan™ Fast Advanced Master Mix (Applied Biosystems, ThermoFisher Scientific) and TaqMan probes. qPCR reactions were carried out in a 10 µL volume, which included 30 ng of cDNA, 5 µL of TaqMan Fast Advanced PCR Master Mix, and 0.5 µL of the appropriate TaqMan probes (20×). All reactions were performed on the 7900HT Fast Real-Time PCR System (Applied Biosystems, ThermoFisher Scientific) under the following conditions: 20 s at 95 °C followed by 40 cycles of 3 s at 95 °C and 30 s at 60 °C. Data were analyzed using SDS 2.3 software and Data Assist (Applied Biosystems, ThermoFisher Scientific). Fold change values (RQ) were calculated using the 2^−ΔΔCT^ method, with *p*-values < 0.05 considered statistically significant.

### 4.9. mRNA Expression Analysis

mRNA expression analysis was performed for the following genes: *CD79*, *Ly6d*, *H2-BMb2*, *Pld4*, *Ffar1*, *Scg2*, *Siglec1*, and *Ctsh*. Quantitative analysis of mRNA expression was performed using RT-qPCR with TaqMan probes for: *Polr2a* (Mm00839493_m1), *CD79* (Mm00434143_m1), *Ly6d* (Mm00521959_m1), *H2-BMb2* (Mm00783707_s1), *Pld4* (Mm00626861_m1), *Ffar1* (Mm00809442_s1), *Scg2* (Mm04207690_m1), *Siglec1* (Mm00556586_m1), and *Ctsh* (Mm00514455_m1) (Life Technologies, ThermoFisher Scientific, MA, USA). Gene expression levels were normalized to RNA polymerase II subunit A (Polr2a).

### 4.10. Statistical Analysis

Continuous variables were subjected to the Shapiro–Wilk test to confirm the normality of their distribution. Based on the results, any differences between the two groups were evaluated using a two-sided independent Student’s *t* test, if the data were normally distributed, or the Mann–Whitney U test if they were not. Differences were considered significant for *p*-values < 0.05. A heatmap of the results was plotted using pheatmap R package (version 0.7.7). The sample size was estimated based on a moderate-to-strong effect size (>0.5 SD) for the primary research questions, aiming for a statistical power of 80% and an alpha level of 0.05. This calculation was specifically applied to the main comparison of KO_MOG vs. WT. The remaining comparisons were exploratory and intended to verify the initial observations: no separate sample size calculations were conducted for them.

## 5. Conclusions

Mice were immunized to induce a response of CD4+ T lymphocytes following stimulation with MOG myelin antigen. Our findings reveal significant differences between CD4+ T cells from MOG_35-55_-stimulated KO mice and WT cells with regard to mRNA expression profile, but not the miRNA profile. MOG_35-55_ antigen stimulation resulted in an increase in *Ffar1* and *Scg2* gene expression in CD4+ T cells from miR-155 KO mice compared to those from WT mice. In addition, these genes were found to be upregulated by real-time PCR using Taqman probes. The increased expression of these genes in the miR-155-deletion mouse group may indicate an alternative mechanism in the autoimmune response.

These results appear to be promising as a potential therapeutic target; however, this needs to be confirmed in a larger study group.

## Figures and Tables

**Figure 1 ijms-25-11355-f001:**
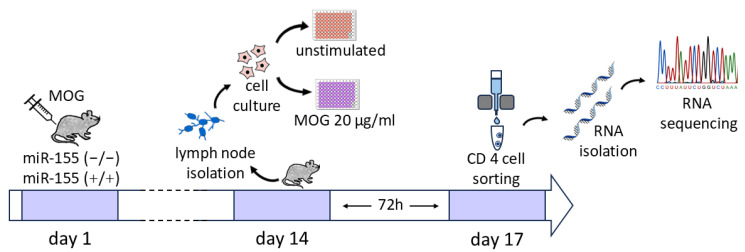
Scheme of the study.

**Figure 2 ijms-25-11355-f002:**
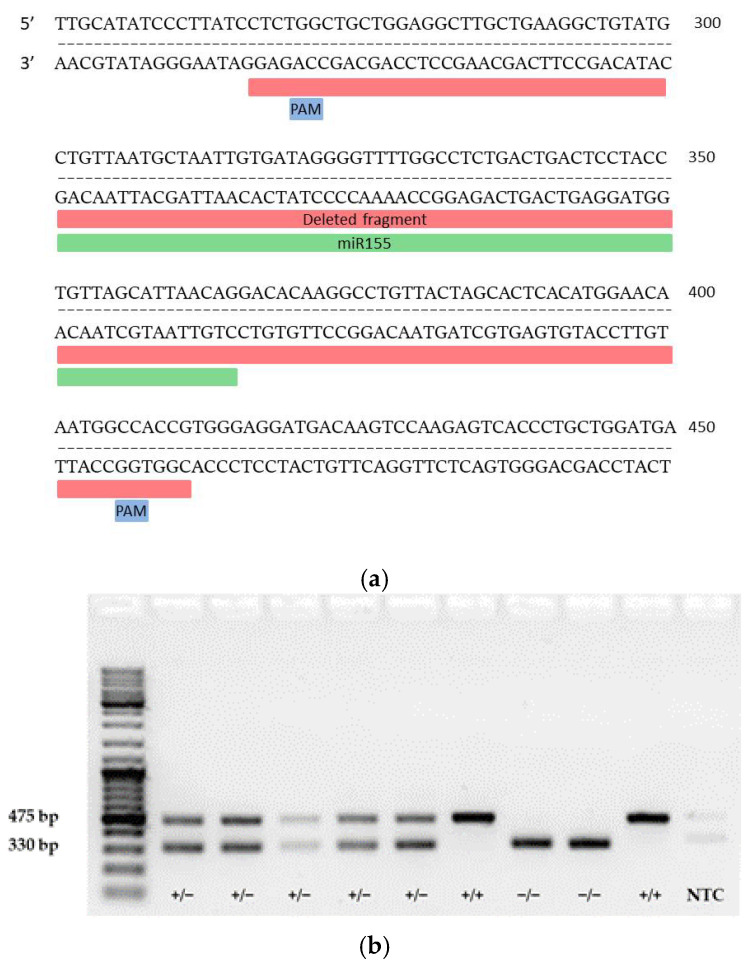
Mutagenesis strategy for generating miRNA-155-deficient mice. (**a**) Diagram showing the miR-155 genomic locus (highlighted in yellow) and the entire region targeted for deletion (highlighted in red). (**b**) Representative PCR genotyping results using described primer pairs to distinguish WT (+/+), heterozygous (+/−), and miR-155 KO mice (−/−). (**c**) Comparison of positive droplet identification (shown in blue) using QuantaSoft Software, version 1.7. On the top left: Wells from A01 to C01 display the presence of miR-155-3p in splenocytes (A01: WT spleen; B01: heterozygous, C01: KO-miR-155) G01: NTC for miR-155-3p. Respectively, on the top right, wells from D01 to F01, include miR-155-5p results in the same order, with H01 as the NTC for miR-155-5p. The graphs below illustrate quantitative copy numbers of miRNA-155-3p and miRNA-155-5p in splenocytes from WT, heterozygous, and miR-155 KO mice and NTC.

**Figure 3 ijms-25-11355-f003:**
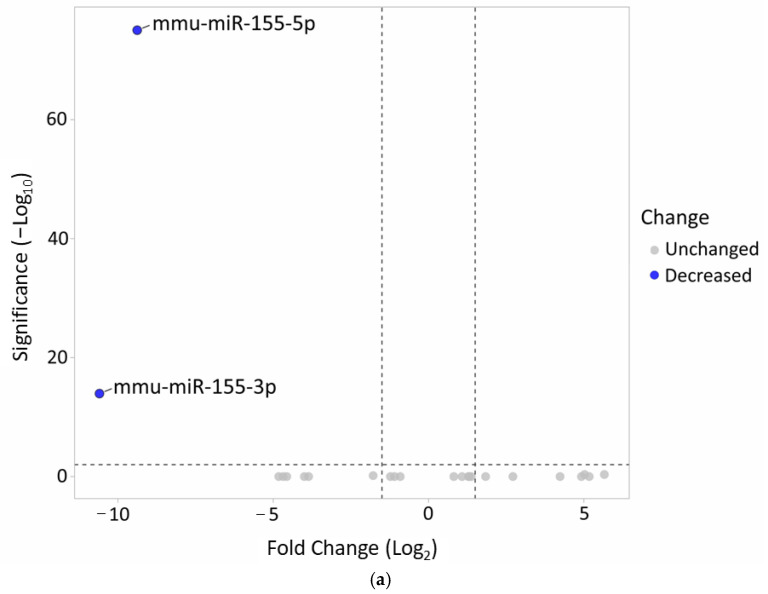
(**a**) Confirmation of the successful deletion of miR-155 in KO mice. (**b**) The heat map illustrates gene expression patterns in CD4+ T cells from both miR-155-sufficient and -deficient mice (*p* = 0.05). CD4+ T cells from KO mice were re-stimulated in vitro with the MOG_35-55_ antigen. Rows are centered; unit variance scaling is applied to rows. Both rows and columns are clustered using correlation distance and average linkage (39 rows, 4 columns). (**c**) Volcano plot highlights the differences in gene expression between MOG-stimulated miR-155-deficient CD4+ T cells and WT CD4+ T unstimulated cells.

**Figure 4 ijms-25-11355-f004:**
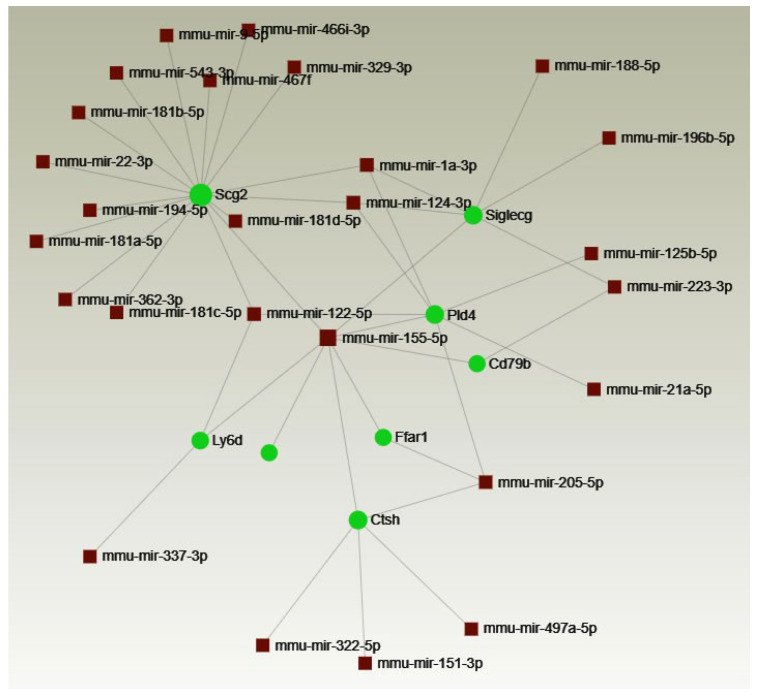
Interactions between genes upregulated in CD4+ T cells from MOG-stimulated miR-155 KO mice and miRNAs. Network generated by miRNet.

**Figure 5 ijms-25-11355-f005:**
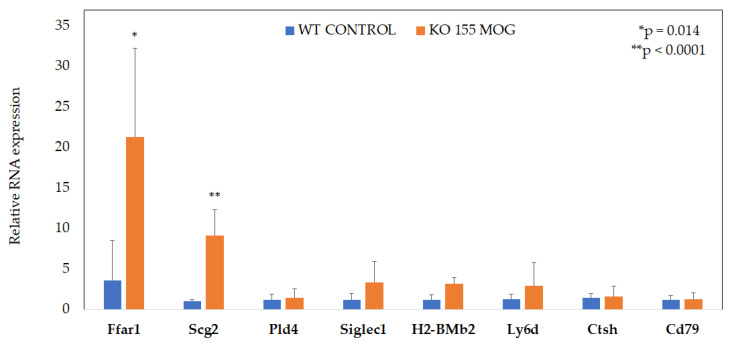
Increased *Ffar1* and *Scg2* expression in MOG stimulated miR-155-deficient CD4+ T cells. Data represent the mean ± SD from three independent experiments, with each variant including 4–5 mice. * *p* = 0.014 or ** *p* < 0.0001.

**Figure 6 ijms-25-11355-f006:**
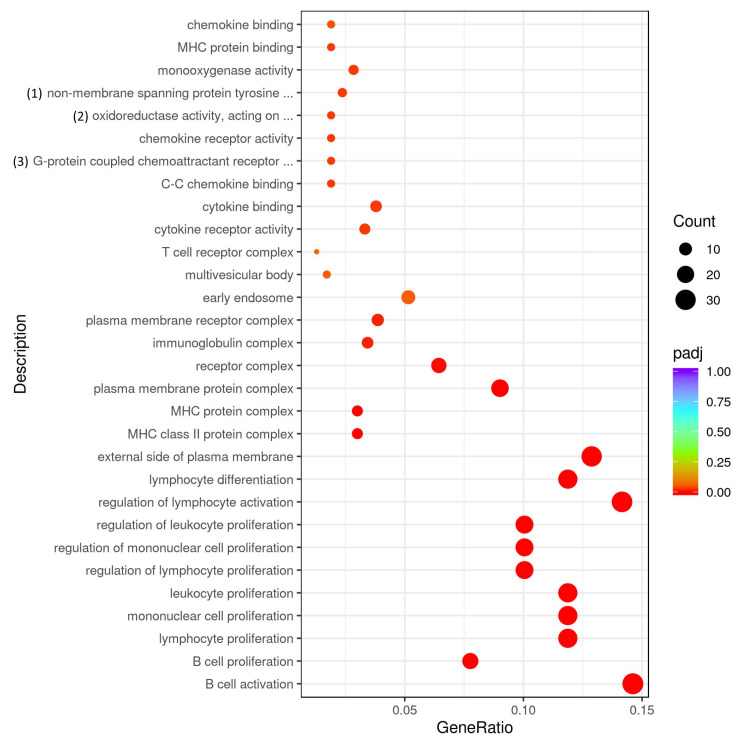
GO analysis of gene expression in MOG restimulated miR-155-deficient CD4+ T cells compared to CD4+ T cells from control mice. Full description of processes: (1) non-membrane spanning protein tyrosine kinase activity; (2) oxidoreductase activity, acting on paired donors, with incorporation or reduction of molecular oxygen, NAD(P)H as one donor, and incorporation of one atom of oxygen; (3) G-protein coupled chemoattractant receptor activity.

## Data Availability

Data is contained within the article or Appendix A.

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
