# Peer review of "The Role of miR-155 in Modulating Gene Expression in CD4+ T Cells: Insights into Alternative Immune Pathways in Autoimmune Encephalomyelitis"

_ijms, 2024, doi:10.3390/ijms252111355_

Round 1

Reviewer 1 Report

Comments and Suggestions for Authors

Your manuscript looks interesting but some suggestions are recommended:

1. Some abbreviation in the introduction must be clarified in their first mention.

2. It is recommended to leave space between paragraphs.

3. Regarding the Institutional Review Board Statement, the dates are conflicting and mysterious. Are these general codes for all proposals involving animals or specific for this study? please clarify.

4. Regarding genotyping, did you design the primer sequence or picked it from already published literature?

5. Why you extracted the DNA not the RNA from the mice if you are looking for the absence of miR-155?

6. Why you didn't perform any histopathological examination for the isolated lymph nodes to ensure the induction?

7. Did you perform any statistical sample size calculations for the used sample number?

8. How did the authors ensure the normality of the results?

9. I suggest adding heading about statistical analyses in methodology section.

10. the results section is clearly presented.

10. The discussion section is comprehensive.

4. 

Author Response

Response to Reviewer 1 Comments

Thank you very much for taking the time to review this manuscript. Please find the detailed responses below and the corresponding corrections in track changes in the re-submitted files.
In order to maintain clarity and consistency. Additional changes have been made:
- Abstract was reconstructed according to Reviewer 2’s suggestions: “Please shorten the background description, and summarize the major results obtained in this study”. The Abstract with corrections is implemented in the manuscript.
- The manuscript was sent to the Centre for Language Teaching at the Medical University of Lodz for linguistic correction. In addition, the order of the keywords has been changed after correction by the Centre for Language Teaching at the Medical University of Lodz.
- 4.9. mRNA expression analysis in the Materials and methods section has been shortened because the method is described in 4.8.
- The supplementary material consisting of mRNA and miRNA sequencing data with
p-value < 0.05 has been added to the manuscript.
- According to Reviewer 2’s suggestions study limitation paragraph has been added into Discussion section and new section Conclusion.

Point-by-point response to Comments and Suggestions for Authors

Comments 1: Some abbreviation in the introduction must be clarified in their first mention.
Response 1: Thank you for pointing this out. According to the Your suggestion abbreviations have been clarified in their first mention in the Abstract and Introduction sections:

  • “knockout mice (KO)”, p. 1, Abstract, L 23.
  • “Signal Transducer and Activator of Transcription 3 (STAT3), and Basic Leucine Zipper ATF-Like Transcription Factor (BATF) ”, p. 1, Introduction, L 41.
  • “Interleukin 17 (IL-17)", p. 1, Introduction, L 42.

Comments 2: It is recommended to leave space between paragraphs.
Response 2: Thank you for pointing this out. According to the Your suggestion we added additional spaces before following sections: Results, Discussion, Materials and Methods and References.

Comments 3: Regarding the Institutional Review Board Statement, the dates are conflicting and mysterious. Are these general codes for all proposals involving animals or specific for this study? please clarify.
Response 3: We appreciate your comment. Our explanation to this issue is:

The approval of the Local Ethical Committee at the Medical University of Lodz was given for the purposes of this project. It considered: the type of study, the species and the number of mice. The first approval from the Local Ethical Committee (69/LB06/2015) covered the period February 2016-January 2019 and was extended until January 2023. Due to the prolonged duration of the project, a new approval was needed in order to conduct further animal studies in this project. The second consent from the Local Ethical Committee (10/LB 261/2023), was issued for the period from February 2023 to January 2026.

In addition, due to the fact that genetically modified mice were used in the research, additional approval was needed from the Ministry of the Environment, issued for an indefinite period of time (DOP-GMO.431.226.2017, decision no 175/2017). The current approvals are public and published on following websites:

  • ALURES – ANIMAL USE REPORTING - EU SYSTEM,
    Approval of the Ethics Committee (EC NTS/RA identifier: NTS-PL-659925)
    https://webgate.ec.europa.eu/envdataportal/web/resources/alures/submission/nts/list?filter=W3sia2V5IjoiY291bnRyeUNvZGUiLCJ2YWx1ZSI6InBsIiwiY29tcGFyaXNvbiI6IkVRVUFMIn0seyJrZXkiOiJ0aXRsZSIsInZhbHVlIjoicG9zenVraXdhbmllICIsImNvbXBhcmlzb24iOiJFUVVBTCJ9LHsia2V5Ijoic3BlY2llcyIsInZhbHVlIjpbIkExIl0sImNvbXBhcmlzb24iOiJFUVVBTFNfQU5ZX0lOX1RIRV9MSVNUIn0seyJrZXkiOiJwdWJsaWNhdGlvblllYXIiLCJ2YWx1ZSI6IjIwMjMiLCJjb21wYXJpc29uIjoiRVFVQUwifSx7ImtleSI6ImV1U3VibWlzc2lvbiIsInZhbHVlIjoieWVzIiwiY29tcGFyaXNvbiI6IkVRVUFMIn1d&draw=0&countryCode=pl&title=poszukiwanie+&species=A1&publicationYear=2023&euSubmission=yes&length=10&orderDirection=desc&search=true
  • Polish Ministry of Climate and Environment (GMO consent)
    https://gmo.klimat.gov.pl/registries/1/cards/1419

Comments 4: Regarding genotyping, did you design the primer sequence or picked it from already published literature?
Response 4: Thank you for this question. Let me clarify. I did not design the primer sequences for genotyping. I received the sequences from the Genome Engineering Facility at the International Institute of Molecular and Cell Biology in Warsaw, the facility which generated miR-155 knockout mice. Primer were designed using SnapGene program and their specificity was verified using BLAST software.

Comments 5: Why you extracted the DNA not the RNA from the mice if you are looking for the absence of miR-155?
Response 5: Thank you for this question. To confirm the miR-155 deletion in the genetic material of miR-155 knockout mice, a two-step verification at the DNA and RNA level was employed. The results of DNA analysis using primers for genotyping are presented as an agarose gel electrophoresis on Fig. 2b in the Materials and Methods section.
RNA was analysed to confirm the miR-155 deletion using TaqMan FAM-labeled fluorescent probes for miR-155-3p or miR-155-5p (ThermoFisher Scientific) by ddPCR method presented on Fig. 2c in the Materials and Methods section.

Comments 6: Why you didn't perform any histopathological examination for the isolated lymph nodes to ensure the induction?
Response 6: Thank you for this question. This step was not planned in the project. However, your suggestion is very valuable and we will consider it in future studies.

Comments 7: Did you perform any statistical sample size calculations for the used sample number?
Response 7: Thank you for this question. We performed a sample size estimation based on a moderate-to-strong effect size (>0.5 SD) for the primary research questions, aiming for a statistical power of 80% and an alpha level of 0.05. This calculation was specifically applied to the main comparison of KO_MOG vs. WT. The remaining comparisons were exploratory and intended to verify the initial observations, and no separate sample size calculations were conducted for those exploratory analyses.

Comments 8: How did the authors ensure the normality of the results?
Response 8: Thank you for pointing this out. In order to verify the normality of the data, we used the Shapiro–Wilk test. If the Shapiro-Wilk test indicated that the data were normally distributed, we applied a two-sided independent Student’s t-test to compare the continuous variables between the two groups. If the assumption of normality was violated, we used the Mann–Whitney U test. Statistical significance was determined by p-values < 0.05.

Comments 9: I suggest adding heading about statistical analyses in methodology section.
Response 9: Thank you for pointing this out. According to the Your suggestion Statistical Analysis section was added as the last paragraph in Materials and Methods section p. 14 Materials and Methods.

Comments 10: the results section is clearly presented.
Response 10: Thank you for this statement.

Comments 10: The discussion section is comprehensive.
Response 10: Thank you for this statement.

Reviewer 2 Report

Comments and Suggestions for Authors

Reviewer’s Comments and Suggestions for Authors

 Journal: Int. J. Mol. Sci., MDPI

Manuscript ID: ijms-3211427

Type: Article

Title: The Role of miR-155 in Modulating Gene Expression in CD4+ T Cells: Insights into Alternative Immune Pathways in Autoimmune Encephalomyelitis

Authors: Maria Cichalewska-Studzinska*, Jacek Szymanski, Emilia Stec-Martyna, Ewelina Perdas, Mirosława Studznńska, Hanna Jerczynska, Dominika ulczycka-Wojdala, Robert Stawski and Marcin P Mycko

The authors of the manuscript Manuscript ID: ijms-3211427 performed mRNA sequencing (mRNA-seq) to profile the transcriptome and quantify gene expression levels in immunized mice. By using miR-155 knockout mice, the authors observed differences in mRNA gene expression in CD4+ T cells in response to immunization with the antigen myelin oligodendrocyte glycoprotein (MOG). The increased expression of Ffar1 (free fatty acid receptor 1) and Scg2 (Secretogranin-2) genes in CD4+ T cells from miR-155-deficient mice after MOG re-stimulation were examples of alternative responses to antigen stimulation. These results may provide deeper insights into CD4+ T cell activation, although further analysis are required.

Many essential information regarding the experiments of this study was missing in the Materials and Methods section, which should be clarified. Moreover, the Results, particularly concerning the mRNA sequencing were presented preliminary. Therefore, this manuscript in its current form can not be recommended for publication.

Major revisions 

1. The Abstract should be reconstructed. Please shorten the background description, and summarize the major results obtained in this study.

2. In the Materials and Methods section, many essential information regarding the experiments of this study was missing, for example,

(1) No reference or source of some methods used in this study was cited, such as the CRISPR/Cas9 method, Benjamini and Hochberg approach, etc. Please clarify.

(2) Did the authors provide the document approved by ethics committees for the animal experiment?.

(3) How many mice did the authors use in this study? How many groups of the mice? Male or female mice? Please provide the detailed information.

(4) How long did the cells were stimulated with MOG35-55 (20 µ g/ml)? 

(5) How many RNA samples were extracted from each group?

(6) The source information of some equipment used in this study was missing. Please clarify.

(7) What was the genome website?

(8) Please add website and accessed date (day month year) of the the software used in this study.

(9) In the “4.6. mRNA sequencing”, “Eight RNA samples were rigorously assessed”. In the “4.7. miRNA sequencing”, “Twelve RNA samples (three each from WT, WT_MOG, KO, and KO_MOG groups) were thoroughly evaluated”. Please explain the difference between the sample numbers.

(10) How were small RNA molecules were isolated? Please clarify.

3.  In the Results section, for example:

(11) The Results of the mRNA sequencing were very preliminary. Please provide the detailed information of how many genes were significantly up-regulated or down-regulated in the CD4+ T cells after the MOG restimulation. 

(12) Regarding the To confirm the successful deletion of the targeted genomic sequence, PCR was performed”, did the authors sequence the amplified PCR products? 

(13) Concerning the “The results revealed diverse molecular interactions in response to antigen stimulation between the samples”, please analyze the data in more details.

(14) Regarding the two genes, Ffar1 and Scg2, showed increased expression, what were fold changes of these genes? 

(15) Figure 3: please add the figure legend.

4. The conclusion was missing. Please summarize the major results obtained in this study.

5. The authors may describe the limitations of this study.

Minor revisions

(16) Abbreviations and acronyms are typically defined the first time the term is used within the abstract and again in the main text and then used throughout the remainder of the manuscript. Please consider adhering to this convention, and check throughout the manuscript.

(17) Line 13: please check the Correspondence:Correspondence:

(18) Line 58: please change “Research indicates” to “Research has indicated that”.

(19) Line 143: please change “p = 0,05” to “p = 0.05”, and check the similar issue throughout the manuscript.

(20) Line 167: please change *p=0.01 or **p<0.0001 to *p = 0.01 or **p < 0.0001, and check the similar issue throughout the manuscript.

(21) Concerning the (RT qPCR) and RT-qPCR, please be consistent.

(22) Lines 212-213: please rephrase the sentence.

(23) Lines 265: please change the 1 µl of the resulting DNA” to “A 1 µl of the resulting DNA”.

(24) Lines 346: please change the “method [33], [34].” to “method [33,34].”, and check the similar issue throughout the manuscript.

(25) Please change “minutes” to “min”, and hours to h throughout the manuscript.

(26) Please format the References according to the guidance of the journal to authors.

(27) There were some English language issues in this manuscript. Please carefully and extensively amend throughout the manuscript.

Comments on the Quality of English Language

Moderate editing of English language required.

Author Response

Response to Reviewer 2 Comments

Thank you very much for taking the time to review this manuscript. Please find the detailed responses below and the corresponding corrections in track changes in the re-submitted files.

In order to maintain clarity and consistency. Additional changes have been made:

- The manuscript was sent to the Centre for Language Teaching at the Medical University of Lodz for linguistic correction. In addition, the order of the keywords has been changed after correction by the Centre for Language Teaching at the Medical University of Lodz.

- 4.9. mRNA expression analysis in the Materials and methods section has been shortened because the method is described in 4.8.

- The supplementary material consisting of mRNA and miRNA sequencing data with

p-value < 0.05 has been added to the manuscript.

Point-by-point response to Comments and Suggestions for Authors

Major revisions 

Comments 1: The Abstract should be reconstructed. Please shorten the background description, and summarize the major results obtained in this study.
Response 1: Thank you for this comment. The Abstract has been reconstructed as below:
CD4+ T cells are considered the main orchestrators of autoimmune diseases. Its disruptive effect on CD4+ T cell differentiation and the imbalance between T helper cell populations, can be most accurately determined using experimental autoimmune encephalomyelitis (EAE) as an animal model of multiple sclerosis (MS). One epigenetic factors known to promote autoimmune inflammation is miRNA-155 (miR-155), which is significantly upregulated in inflammatory T cells.
The aim of the present study was to profile the transcriptome of immunized mice and determine their gene expression levels based on mRNA and miRNA sequencing. No statistically significant differences in miRNA profile were observed; however, substantial changes in gene expression between miRNA-155 knockout (KO) mice and WT were noted. In miR-155 KO mice, mRNA ex-pression in CD4+ T cells changed in response to immunization with the myeloid antigen MOG35-55. After restimulation with MOG35-55, increased Ffar1 (free fatty acid receptor 1) and Scg2 (secretogranin-2) expression were noted in the CD4+ T cells of miR-155-deficient mice; this is an example of an alternative response to antigen stimulation
.

2. In the Materials and Methods section, many essential information regarding the experiments of this study was missing, for example,

Comments 2.1: No reference or source of some methods used in this study was cited, such as the CRISPR/Cas9 method, Benjamini and Hochberg approach, etc. Please clarify.
Response 2.1: Thank you for highlighting this. Appropriate reference was added to the manuscript Reference number 35. p. 13, L 378

Comments 2.2: Did the authors provide the document approved by ethics committees for the animal experiment?
Response 2.2: Thank you for highlighting this issue. Initially we attached the ethical statement in the section entitled Institutional Review Board Statement. However, after Assistant Editor suggestion following statement was pasted at the end of the paragraph 4.1 Mice in the paragraph 4. Materials and Methods:
“……The animal study protocol was approved by the Local Ethics Committee at the Medical University of Lodz (decisions 69/ŁB06/2015 -7th of December 2015 and 10/LB 261/2023 – 6th of February 2023) for studies involving animals. Research conducted on GMO mice was also approved by the Polish Ministry of the Environment (DOP-GMO.431.226.2017 decision no 175/2017).”
Scan of approvals of the Ethics Committee for years 2015-2023 are in enclosed document in the “Non-published Material” on website. The current approvals are public and published on following websites:

1)    ALURES – ANIMAL USE REPORTING - EU SYSTEM,
Approval of the Ethics Committee (EC NTS/RA identifier: NTS-PL-659925)
https://webgate.ec.europa.eu/envdataportal/web/resources/alures/submission/nts/list?filter=W3sia2V5IjoiY291bnRyeUNvZGUiLCJ2YWx1ZSI6InBsIiwiY29tcGFyaXNvbiI6IkVRVUFMIn0seyJrZXkiOiJ0aXRsZSIsInZhbHVlIjoicG9zenVraXdhbmllICIsImNvbXBhcmlzb24iOiJFUVVBTCJ9LHsia2V5Ijoic3BlY2llcyIsInZhbHVlIjpbIkExIl0sImNvbXBhcmlzb24iOiJFUVVBTFNfQU5ZX0lOX1RIRV9MSVNUIn0seyJrZXkiOiJwdWJsaWNhdGlvblllYXIiLCJ2YWx1ZSI6IjIwMjMiLCJjb21wYXJpc29uIjoiRVFVQUwifSx7ImtleSI6ImV1U3VibWlzc2lvbiIsInZhbHVlIjoieWVzIiwiY29tcGFyaXNvbiI6IkVRVUFMIn1d&draw=0&countryCode=pl&title=poszukiwanie+&species=A1&publicationYear=2023&euSubmission=yes&length=10&orderDirection=desc&search=true

2)            Polish Ministry of Climate and Environment (GMO consent)
https://gmo.klimat.gov.pl/registries/1/cards/1419

Comments 2.3: How many mice did the authors use in this study? How many groups of the mice? Male or female mice? Please provide the detailed information
Response 2.3: Thank you for your questions. In all experiments we used 47 mice, 24 miR-155 knockout mice (KO) and 23 miR-155 sufficient mice (one mice died during experiment). We used female mice aged between 8-12 weeks.

Comments 2.4: How long did the cells were stimulated with MOG35-55 (20 µ g/ml)? 
Response 2.4: Thank you for your question. Cells were stimulated with MOG35-55 or left unstimulated for 72 h. It might have not be clearly described in Materials and Methods section 4.3 Immunization and lymph nodes cells culture. Hence, rearrangement of this section was performed from “…The cells were either stimulated with MOG35-55 (20 µg/ml) or left unstimulated. Cell viability was assessed using propidium iodide. After 72 hours, cells were sorted using indirect magnetic sorting with the CD4+ T Cells Isolation Kit II (Miltenyi Biotech, Germany) according to the manufacturer’s instructions.” to “….”The cells were cultured for 72 h, either stimulated with MOG35-55 (20 µg/ml) or left unstimulated. Cell viability was assessed using propidium iodide (CytoFLEX, Beckman Coulter, USA). Subsequently, cells were sorted using indirect magnetic sorting with the CD4+ T Cells Isolation Kit II (Miltenyi Biotech, Germany) according to the manufacturer’s instructions.”

Comments 2.5: How many RNA samples were extracted from each group?
Response 2.5: Thank you for your question. For miRNA sequencing 6 RNA samples were isolated from 24 KO mice and 6 samples from 23 WT mice. In total, 12 RNA samples (three from each group: WT, WT_MOG, KO, and KO_MOG) were isolated. Each RNA samples was isolated from lymphocytes derived from three or four mice respectively.

Comments 2.6: The source information of some equipment used in this study was missing. Please clarify.
Response 2.6: Thank you for pointing this out. We apologize for our omission. Missing equipment details were added to the manuscript in Materials and Methods section:
- 4.2 Genotyping “…PCR was performed on the Gene Amp PCR System 9700 (Applied Biosystems, ThermoFisher Scientific)….” p. 11, L 297-298.
- 4.3 Immunization and lymph nodes cell culture “…Cell viability was assessed using propidium iodide (CytoFLEX, Beckman Coulter, USA)….” p. 12, L 312-313.
- 4.8. Reverse transcription and real-time quantitative PCR (RT qPCR) “…Reverse transcription was performed on the Gene Amp PCR System 9700 using …” p. 14 L 424.
Procedures described in sections 4.6. mRNA sequencing and 4.7. miRNA sequencing were performed at Novogene Europe, Cambridge, United Kingdom.

Comments 2.7: What was “the genome website”?
Response 2.7: Thank you for this question. Reference genome (ensembl_mus_musculus_grcm38_p6_gca_000001635_8) and gene model annotation files were directly downloaded from Ensemble.

Comments 2.8: Please add website and accessed date (day month year) of the the software used in this study.
Response 2.8: Thank you pointing this out. Interactions between genes and miRNAs shown in Figure 4 were generated using miRNET 2.0 software (http://www.mirnet.ca/, Accessed 12 June 2024). This information was added to 2.4 miRNA-155 has multiple gene targets in Results section.

Comments 2.9: In the “4.6. mRNA sequencing”, “Eight RNA samples were rigorously assessed”. In the “4.7. miRNA sequencing”, “Twelve RNA samples (three each from WT, WT_MOG, KO, and KO_MOG groups) were thoroughly evaluated”. Please explain the difference between the sample numbers.
Response 2.9: Thank you for this question. According to the project plan we were focused on finding changes in miRNA profile between KO and WT samples after MOG stimulation (12 samples three from each: WT, WT_MOG, KO, and KO_MOG groups). mRNA sequencing was additional step (out of project plan). Only in 8 samples (two from each: WT, WT_MOG, KO, and KO_MOG) there was enough RNA for mRNA sequencing. Nevertheless, it provided a valuable feedback. That is why there is a discrepancy between sample number in both sequencing.

Comments 2.10: How were small RNA molecules were isolated? Please clarify.
Response 2.10: Thank you for this question. RNA samples were isolated with miRNeasy Mini Kit (Qiagen). According to the manual, in this protocol total RNA (including small RNAs) is isolated. We did not perform any additional miRNA enrichment procedures.

3.  In the Results section, for example:

Comments 3.11: The Results of the mRNA sequencing were very preliminary. Please provide the detailed information of how many genes were significantly up-regulated or down-regulated in the CD4+ T cells after the MOG restimulation. 
Response 3.11: Thank you for this question. Data from mRNA and miRNA sequencing of changes between KO MOG and WT was attached in supplementary materials. Differential expression analysis between conditions or groups with p adj. < 0,05. In CD4+ T cells after MOG restimulation from miR-155 deficient mice to control WT we observed upregulation of 23 genes and 16 genes were down-regulated.

Comments 3.12: Regarding the “To confirm the successful deletion of the targeted genomic sequence, PCR was performed”, did the authors sequence the amplified PCR products? 
Response 3.12: Thank you for pointing this out. It is a good idea to sequence the PCR product, which would ascertain us with the successful deletion of miR-155 sequence. However, we did not sequence PCR product, we compared the sizes of the product on electrophoresis agarose gel. We will introduce this step in the future projects.

Comments 3.13: Concerning the “The results revealed diverse molecular interactions in response to antigen stimulation between the samples”, please analyze the data in more details.
Response 3.13: Thank you for raising the issue. In this project we focused on indication of changes in gene expression in lymphocytes CD4+ T cells in the absence of miR-155. Mechanisms that regulate molecular interaction are under consideration for future project so therefore your comment is a valuable input. We raised this issue in study limitation.

Comments 3.14: Regarding the “two genes, Ffar1 and Scg2, showed increased expression”, what were fold changes of these genes? 
Response 3.14: Thank you for your question. The expression of Ffar1 and Scg2 genes in miR-155-deficient CD4+ T cells stimulated with MOG was increased by approximately 6-fold and 9-fold, respectively, compared to WT cells.

Comments 3.15: Figure 3: please add the figure legend.
Response 3.15: Thank you for your comment. To the legend of The heatmap (figure 3b) following information was added: “Rows are centered: unit variance scaling is applied to rows. Both rows and columns are clustered using correlation distance and average linkage. 39 rows, four columns.” p. 7, L 151-153

Comments 4: The conclusion was missing. Please summarize the major results obtained in this study.
Response 4: Thank you for your suggestion. Section Conclusions was added to the main manuscript as a 5th section: “Immunization of mice to induce a response of CD4+ T lymphocytes following stimulation with MOG myelin antigen showed significant differences between CD4+ T cells KO MOG35-55 stimulated and WT in the mRNA expression profile but not  the miRNA profile. MOG35-55 antigen stimulation resulted in an  increase in Ffar1 and Scg2 gene expression in CD4+ T cells from miR-155 KO mice compared to those from WT mice. In addition, these genes were found to be upregulated by real-time PCR using Taqman probes. The in-creased expression of these genes in the miR-155 deletion mouse group may indicate an alternative mechanism in the autoimmune response.

These results appear to be promising as a potential therapeutic target; however, this needs to be confirmed in a larger group study..”, p. 15, L 454-464.

Comments 5: The authors may describe the limitations of this study.
Response 5: Thank you for your comment. We introduced the following paragraph at the end of the Discussion section: “There are a few limitations of the study. Although the findings indicate Ffar1 and Scg2 to be upregulated in miR-155-deficient CD4+ T cells, further investigation is necessary to clarify the functional significance of these gene responses and their potential roles in immune regulation. In addition, the research primarily focuses on immune responses in peripheral lymphoid tissues, potentially overlooking crucial processes occurring in the CNS, where EAE has significant implications this may hinder a comprehensive under-standing of disease progression within the CNS. Additionally, the use of a murine model may not entirely reflect the pathophysiology of human MS, which could impact the broader applicability of the findings. Validation with a larger sample number is also recommended to enhance the robustness of the conclusions.” p. 11, L 257-266.

Minor revisions

Comments 16: Abbreviations and acronyms are typically defined the first time the term is used within the abstract and again in the main text and then used throughout the remainder of the manuscript. Please consider adhering to this convention, and check throughout the manuscript.
Response 16: Thank you for your valuable comment. All Abbreviations and acronyms have been changed throughout the manuscript

Comments 17: Line 13: please check the “Correspondence:Correspondence:”

Response 17: Thank you for pointing this out. We agree with this comment. The second word “Correspondence:” has been deleted. We apologize for our inattention.

Comments 18: Line 58: please change “Research indicates” to “Research has indicated that”.

Response 18: Thank you for pointing this out. We agree with this comment. Therefore, according to the Your suggestion: Research indicates has been changed to Research has indicated that, p. 2, Introduction, L 60.

Comments 19: Line 143: please change “p = 0,05” to “p = 0.05”, and check the similar issue throughout the manuscript.
Response 19: Thank you for pointing this out. Therefore, according to the Your suggestion: p = 0,05 has been changed to p = 0.05 and this issue has been checked throughout the manuscript.

Comments 20: Line 167: please change “*p=0.01 or **p<0.0001” to “*= 0.01 or **< 0.0001”, and check the similar issue throughout the manuscript.
Response 20: Thank you for pointing this out. Therefore, according to the Reviewer’s suggestion: “*p=0.01 or **p<0.0001 has been changed to “*p = 0.01 or **p < 0.0001”, p. 8, Results, L 172 and these issues has been checked throughout the manuscript.

Comments 21: Concerning the “(RT qPCR)” and “RT-qPCR”, please be consistent.
Response 21: Thank you for pointing this out. According to the Reviewer’s suggestion “RT qPCR” has been changed to “RT-qPCR” and it has been checked throughout the manuscript.

Comments 22: Lines 212-213: please rephrase the sentence.
Response 22: Thank you for pointing this out. According to the Reviewer’s suggestion sentence “In our study focused on CD4+ T cells in the periphery, particularly in lymph nodes, rather than those crossing the blood-brain barrier.” has been changed to “Our present study focuses on CD4+ T cells in the peripheral lymph nodes, rather than those that cross the blood-brain barrier., p. 10, Discussion, L 226-227.

Comments 23: Lines 265: please change the “1 µl of the resulting DNA” to “A 1 µl of the resulting DNA”.
Response 23: Thank you for pointing this out. According to the Your suggestion “1 µl of the resulting DNA” has been changed to “A 1 µl sample of the resulting DNA”, p 11, Materials and Methods, L 294.

Comments 24: Lines 346: please change the “method [33], [34].” to “method [33,34].”, and check the similar issue throughout the manuscript.
Response 24
: Thank you for pointing this out. According to the Your suggestion “method [33], [34.]” has been changed to “method [33,34].” and it has been changed throughout the manuscript.

Comments 25: Please change “minutes” to “min”, and “hours” to “h” throughout the manuscript.
Response 25
: Thank you for pointing this out. According to the Your suggestion the names of time units have been changed to their corresponding letter abbreviations in all the manuscript.

Comments 26: Please format the References according to the guidance of the journal to authors.
Response 26: Thank you for pointing this out. According to the Your suggestion we modified Reference section with Mendeley Citation Style dedicated for International Journal of Molecular Sciences.

Comments 27: There were some English language issues in this manuscript. Please carefully and extensively amend throughout the manuscript.
Response 27: Thank you for pointing this out. The manuscript was sent for linguistic correction to the Center for Language Teaching at Medical University of Lodz. Following corrections were introduced in sections:
a) order of Keywords,
b) Results:
- 2.2. Mutagenesis and miR-155 KO mice generation;
- description of Fig. 2;
- 2.3. MOG restimulation induces changes in gene expression profiles in T cells from miRNA-155 sufficient and miRNA-155 deficient mice;
- description of Fig. 3;
- 2.4. miRNA-155 has multiple gene targets;
- 2.5. Ffar1 and Scg2 upregulation in miR-155 deficient CD4+ T cells;
c) Discussion;
d) Materials and Methods section:
- 4.1. Mice;
- 4.2. Genotyping;
- 4.3. Immunization and lymph nodes cells culture;
- 4.5. Absolute gene expression with digital quantitative PCR;
- 4.6. mRNA sequencing;
- 4.6.1. Library construction, quality control, and sequencing;
- 4.6.2. Bioinformatics analysis;
- 4.7.1. Quality control;
- 4.7.2. Library construction, quality control and sequencing;
- 4.7.3. Bioinformatics analysis for miRNA;
- 4.10. Statistical Analysis;
e) Conclusions.

Reviewer 3 Report

Comments and Suggestions for Authors

Aim, significance and novelty

This is a novel pioneering explorative work.

This study aimed to sequence mRNA (mRNA-seq) to profile the transcriptome and quantify gene expression levels in immunized mice. Using miR-155 knockout mice, the study found differences in mRNA gene expression in CD4+ T cells in response to immunization with the myelin antigen MOG35-55, manifested as an increased expression of Ffar1 (free fatty acid receptor 1) and Scg2 (Secretogranin-2) genes in CD4+ T 24 cells  from miR-155-deficient mice after MOG35-55 re-stimulation are examples of alternative responses to antigen stimulation.

The study postulated that the lack of miR-155 may influence the behavior of CD4+ T cells, and thus elucidates the role of this miRNA in regulating autoimmune responses.

This research is expected to provide valuable insights into the mechanisms by which miR-155 modulates immune function and its potential implications for the development of novel therapeutic approaches for autoimmune diseases.

The obtained results may provide new perspectives into CD4+ T cell activation, inspiring other explorative endeavors to navigate in this dark area which is waiting further explorations.

Figures

The clarity and legibility of the figures are low and needs amelioration. Particularly Figure 1, Figure 2 (a and b and first raw of c) and Figure 3 need to be enlarged and ameliorated.

The legends in Figure 5 should be amplified.

Minor corrections

L14, change (are considered as the) to (are considered the)

L15-16, change (provides better) to (provides a better)

L22, change (by using) to (Using)

L22, change (were able to observe) to (could observe)

L27, change (analysis are) to (analysis is)

L37, change (both the initiation and progression of the disease) to (the disease's initiation and progression.)

L44, change (the human disease.) to (human disease.)

L174-175, change (In experimental autoimmune) to (In the experimental autoimmune)

L179, change (their specific contributions have not been thoroughly explored.) to (their specific contributions have yet to be thoroughly explored.)

L186, change (in regulation of) to (in the regulation of)

L187, change (Damage of the myelin) to (Damage to the myelin)

L193, change (for pathogenesis) to (for the pathogenesis)

L206, change (which emphasize) to (which emphasizes)

L209, change ([21] however,) to ([21]. However,)

L218, change (secretion, but it is) to (secretion. However,  it is)

L229, change (in regulation of) to (in the regulation of)

L230, change (Damage of the myelin) to (Damage to the myelin)

L235, change (microglia which) to (microglia, which)

L236, change (for pathogenesis) to (for the pathogenesis)

L238, change (through alternative) to (through an alternative)

Comments on the Quality of English Language

English needs moderate editing

Author Response

Response to Reviewer 3 Comments

Thank you very much for taking the time to review this manuscript. Please find the detailed responses below and the corresponding corrections in track changes in the re-submitted files.

In order to maintain clarity and consistency. Additional changes have been made:

- Abstract was reconstructed according to Reviewer 2’s suggestions: “Please shorten the background description, and summarize the major results obtained in this study”. The Abstract with corrections is implemented in the manuscript.

- The manuscript was sent to the Centre for Language Teaching at the Medical University of Lodz for linguistic correction. In addition, the order of the keywords has been changed after correction by the Centre for Language Teaching at the Medical University of Lodz.

- 4.9. mRNA expression analysis in the Materials and methods section has been shortened because the method is described in 4.8.

- The supplementary material consisting of mRNA and miRNA sequencing data with

p-value < 0.05 has been added to the manuscript.
- According to Reviewer 2’s suggestions study limitation paragraph has been added into Discussion section and new section Conclusion.

3. Point-by-point response to Comments and Suggestions for Authors

Figures

Comment: The clarity and legibility of the figures are low and needs amelioration. Particularly Figure 1, Figure 2 (a and b and first raw of c) and Figure 3 need to be enlarged and ameliorated.

The legends in Figure 5 should be amplified.

Response: Thank you for the comment. According to Your suggestions, we improved the legibility of the figures.

Minor corrections

Comments 1: L14, change (are considered as the) to (are considered the)
Response 2: Thank you for the comment. It has been changed, p. 1, Abstract.

Comments 2: L15-16, change (provides better) to (provides a better)

Response 2: Thank you for pointing this out. After the Abstract reconstruction and Centre for Language Teaching at the Medical University of Lodz linguistic correction, this expression has been deleted.

Comments 3: L22, change (by using) to (Using) and change (were able to observe) to (could observe)

Response 3: Thank you for pointing this out. After the Abstract reconstruction and Centre for Language Teaching at the Medical University of Lodz linguistic correction, this expression has been deleted.

Comments 4: L27, change (analysis are) to (analysis is)

Response 4: Thank you for pointing this out. After the Abstract reconstruction and Centre for Language Teaching at the Medical University of Lodz linguistic correction, this expression has been deleted.

Comments 5: L37, change (both the initiation and progression of the disease) to (the disease's initiation and progression.)

Response 5: Thank you for pointing this out. We agree with this comment. Therefore, according to the Your suggestion: “both the initiation and progression of the disease” has been changed to “the disease's initiation and progression”, p. 1, Introduction, L37.

Comments 6: L44, change (the human disease.) to (human disease.)

Response 6: Thank you for pointing this out. We agree with this comment. Therefore, according to the Your suggestion: “the human disease” has been changed to “human disease”, p. 2, Introduction, L45-46.

Comments 7: L174-175, change (In experimental autoimmune) to (In the experimental autoimmune)

Response 7: Thank you for pointing this out. We agree with this comment. Therefore, according to the Your suggestion: “In experimental autoimmune” has been changed to “In the experimental autoimmune”, p. 9, Discussion, L187.

Comments 8: L179, change (their specific contributions have not been thoroughly explored.) to (their specific contributions have yet to be thoroughly explored.)

Response 8: Thank you for pointing this out. We agree with this comment. Therefore, according to the Your suggestion: “their specific contributions have not been thoroughly explored” has been changed to “their specific contributions have yet to be thoroughly explored”, p. 9, Discussion, L192-193.

Comments 9: L186, change (in regulation of) to (in the regulation of)

Response 9: Thank you for pointing this out. We agree with this comment. Therefore, according to the Your suggestion: “in regulation of” has been changed to “in the regulation of”, p. 10, Discussion, L243.

Comments 10: L187, change (Damage of the myelin) to (Damage to the myelin)

Response 10: Thank you for pointing this out. We agree with this comment. Therefore, according to the Your suggestion: “Damage of the myelin” has been changed to “Damage to the myelin”, p. 10, Discussion, L244.

Comments 11: L193, change (for pathogenesis) to (for the pathogenesis)

Response 11 Thank you for pointing this out. We agree with this comment. Therefore, according to the Your suggestion: “for pathogenesis” has been changed to “for the pathogenesis”, p. 10, Discussion, L250.

Comments 12: L206, change (which emphasize) to (which emphasizes)

Response 12 Thank you for pointing this out. After linguistic correction this expression has been deleted.

Comments 13: L209, change ([21] however,) to ([21]. However,)

Response 13 Thank you for pointing this out. After linguistic correction this expression has been deleted.

Comments 14: L218, change (secretion, but it is) to (secretion. However, it is)

Response 14 Thank you for pointing this out. We agree with this comment. Therefore, according to the Your suggestion: secretion, but it is,” has been changed to “secretion. However, it is”, p. 13, Discussion, L262.

Comments 15: L229, change (in regulation of) to (in the regulation of)

Response 15 Thank you for pointing this out. We agree with this comment. Therefore, according to the Your suggestion: in regulation of” has been changed to “in the regulation of”, p. 13, Discussion, L275.

Comments 16: L230, change (Damage of the myelin) to (Damage to the myelin)

Response 16 Thank you for pointing this out. Therefore, according to the Your suggestion: Damage of the myelin” has been changed to “Damage to the myelin”, p. 13, Discussion, L276.

Comments 17: L235, change (microglia which) to (microglia, which)

Response 17 Thank you for pointing this out. We agree with this comment. Therefore, according to the Your suggestion: microglia which” has been changed to “microglia, which”, p. 13, Discussion, L282.

Comments 18: L236, change (for pathogenesis) to (for the pathogenesis)

Response 18 Thank you for pointing this out. We agree with this comment. Therefore, according to the Your suggestion: for pathogenesis” has been changed to “for the pathogenesis”, p. 13, Discussion, L283.

Comments 19: L238, change (through alternative) to (through an alternative)

Response 19 Thank you for pointing this out. We agree with this comment. Therefore, according to the Reviewer’s suggestion: through alternative” has been changed to “through an alternative”, p. 13, Discussion, L284-285.

Round 2

Reviewer 1 Report

Comments and Suggestions for Authors

Thank you for your responses. I just suggest adding the responses for comments 4 and 7 in the manuscript to be clearer for the readers.

Author Response

(Second Round)

Journal: Int. J. Mol. Sci., MDPI

Manuscript IDijms-3211427

Type: Article

Title: The Role of miR-155 in Modulating Gene Expression in CD4+ T Cells: Insights into Alternative Immune Pathways in Autoimmune Encephalomyelitis

Authors: Maria Cichalewska-Studzinska*, Jacek Szymanski, Emilia Stec-Martyna, Ewelina Perdas, Mirosława Studzinska, Hanna Jerczynska, Dominika Kulczycka-Wojdala, Robert Stawski and Marcin P Mycko

Response to Reviewer 1 Comments

Thank you very much for taking the time to review this manuscript. Please find the detailed responses below and the corresponding corrections in track changes in the resubmitted files. According to Your suggestion: “I just suggest adding the responses for comments 4 and 7 in the manuscript to be clearer for the readers.” the  following changes were introduced. Our responses have been prepared on the basis of the comments and responses from Round 1 (grey color) in order to improve clarity and readability.

(1) Comments 4: Regarding genotyping, did you design the primer sequence or picked it from already published literature?
Response 4: Thank you for this question. Let me clarify. I did not design the primer sequences for genotyping. I received the sequences from the Genome Engineering Facility at the International Institute of Molecular and Cell Biology in Warsaw, the facility which generated miR-155 knockout mice. Primer were designed using SnapGene program and their specificity was verified using BLAST software.

Response 4R2: Thank you for pointing this out. The sentence „The primers were designed using SnapGene program and their specificity was verified using BLAST software.” has been added to 4.2. Genotyping in the Materials and Methods section, p. 11, L300-301.

(2)Comments 7: Did you perform any statistical sample size calculations for the used sample number?
Response 7: Thank you for this question. We performed a sample size estimation based on a moderate-to-strong effect size (>0.5 SD) for the primary research questions, aiming for a statistical power of 80% and an alpha level of 0.05. This calculation was specifically applied to the main comparison of KO_MOG vs. WT. The remaining comparisons were exploratory and intended to verify the initial observations, and no separate sample size calculations were conducted for those exploratory analyses.

Response 7R2: Thank you for pointing this out. The following fragment “The sample size was estimated based on a moderate-to-strong effect size (>0.5 SD) for the primary research questions, aiming for a statistical power of 80% and an alpha level of 0.05. This calculation was specifically applied to the main comparison of KO_MOG vs. WT. The remaining comparisons were exploratory and intended to verify the initial observations: no separate sample size calculations were conducted for them.” has been added to 4.10. Statistical Analysis in the Materials and Methods section, p. 14, L470-475.

Reviewer 2 Report

Comments and Suggestions for Authors

Reviewer’s Comments and Suggestions for Authors

(Second Round)

Journal: Int. J. Mol. Sci., MDPI

Manuscript ID: ijms-3211427

Type: Article

Title: The Role of miR-155 in Modulating Gene Expression in CD4+ T Cells: Insights into Alternative Immune Pathways in Autoimmune Encephalomyelitis

Authors: Maria Cichalewska-Studzinska*, Jacek Szymanski, Emilia Stec-Martyna, Ewelina Perdas, Mirosława Studznńska, Hanna Jerczynska, Dominika ulczycka-Wojdala, Robert Stawski and Marcin P Mycko

The authors have addressed my comments and suggestions for authors. However, there are still some issues that should be clarified in the text, for example:

(1) Comments 2.3: How many mice did the authors use in this study? How many groups of the mice? Male or female mice? Please provide the detailed information
Response 2.3: Thank you for your questions. In all experiments we used 47 mice, 24 miR-155 knockout mice (KO) and 23 miR-155 sufficient mice (one mice died during experiment). We used female mice aged between 8-12 weeks.

The authors should describe this Response in the Materials and Methods section, and point out the line numbers of the corresponding revisions in the revised manuscript.

(2) Comments 2.5: How many RNA samples were extracted from each group?
Response 2.5: Thank you for your question. For miRNA sequencing 6 RNA samples were isolated from 24 KO mice and 6 samples from 23 WT mice. In total, 12 RNA samples (three from each group: WT, WT_MOG, KO, and KO_MOG) were isolated. Each RNA samples was isolated from lymphocytes derived from three or four mice respectively.

The authors should describe this Response in the Materials and Methods section, and point out the line numbers of the corresponding revisions in the revised manuscript.

(3) Comments 2.7: What was “the genome website”?
Response 2.7: Thank you for this question. Reference genome (ensembl_mus_musculus_grcm38_p6_gca_000001635_8) and gene model annotation files were directly downloaded from Ensemble.

The authors should describe this Response in the Materials and Methods section, and point out the line numbers of the corresponding revisions in the revised manuscript.

(4) Comments 2.9: In the “4.6. mRNA sequencing”, “Eight RNA samples were rigorously assessed”. In the “4.7. miRNA sequencing”, “Twelve RNA samples (three each from WT, WT_MOG, KO, and KO_MOG groups) were thoroughly evaluated”. Please explain the difference between the sample numbers.
Response 2.9: Thank you for this question. According to the project plan we were focused on finding changes in miRNA profile between KO and WT samples after MOG stimulation (12 samples three from each: WT, WT_MOG, KO, and KO_MOG groups). mRNA sequencing was additional step (out of project plan). Only in 8 samples (two from each: WT, WT_MOG, KO, and KO_MOG) there was enough RNA for mRNA sequencing. Nevertheless, it provided a valuable feedback. That is why there is a discrepancy between sample number in both sequencing.

The authors should describe this Response in the Materials and Methods section, and point out the line numbers of the corresponding revisions in the revised manuscript.

(5) Comments 3.11: The Results of the mRNA sequencing were very preliminary. Please provide the detailed information of how many genes were significantly up-regulated or down-regulated in the CD4+ T cells after the MOG restimulation.
Response 3.11: Thank you for this question. Data from mRNA and miRNA sequencing of changes between KO MOG and WT was attached in supplementary materials. Differential expression analysis between conditions or groups with p adj. < 0,05. In CD4+ T cells after MOG restimulation from miR-155 deficient mice to control WT we observed upregulation of 23 genes and 16 genes were down-regulated.

The authors should describe this Response in the Results section, and point out the line numbers of the corresponding revisions in the revised manuscript.

(6) Comments 3.12: Regarding the “To confirm the successful deletion of the targeted genomic sequence, PCR was performed”, did the authors sequence the amplified PCR products?
Response 3.12: Thank you for pointing this out. It is a good idea to sequence the PCR product, which would ascertain us with the successful deletion of miR-155 sequence. However, we did not sequence PCR product, we compared the sizes of the product on electrophoresis agarose gel. We will introduce this step in the future projects.

The authors should discuss this Response in the Discussion section, and point out the line numbers of the corresponding revisions in the revised manuscript.

(7) Comments 3.14: Regarding the “two genes, Ffar1 and Scg2, showed increased expression”, what were fold changes of these genes?
Response 3.14: Thank you for your question. The expression of Ffar1 and Scg2 genes in miR-155-deficient CD4+ T cells stimulated with MOG was increased by approximately 6-fold and 9-fold, respectively, compared to WT cells.

The authors should describe this Response in the Results section, and point out the line numbers of the corresponding revisions in the revised manuscript.

Comments on the Quality of English Language

Minor editing of English language required.

Author Response

(Second Round)

Journal: Int. J. Mol. Sci., MDPI

Manuscript IDijms-3211427

Type: Article

Title: The Role of miR-155 in Modulating Gene Expression in CD4+ T Cells: Insights into Alternative Immune Pathways in Autoimmune Encephalomyelitis

Authors: Maria Cichalewska-Studzinska*, Jacek Szymanski, Emilia Stec-Martyna, Ewelina Perdas, Mirosława Studzinska, Hanna Jerczynska, Dominika Kulczycka-Wojdala, Robert Stawski and Marcin P Mycko

Response to Reviewer 2 Comments

Thank You very much for taking the time to review this manuscript. Please find the detailed responses below and the corresponding corrections in track changes in the resubmitted files. Our responses have been prepared on the basis of the comments and responses from Round 1 (grey color) in order to improve clarity and readability. In addition, the manuscript has been sent to the Foreign Language Centre of the Medical University of Lodz for linguistic correction. 

(1) Comments 2.3: How many mice did the authors use in this study? How many groups of the mice? Male or female mice? Please provide the detailed information
Response 2.3: Thank you for your questions. In all experiments we used 47 mice, 24 miR-155 knockout mice (KO) and 23 miR-155 sufficient mice (one mice died during experiment). We used female mice aged between 8-12 weeks.

The authors should describe this Response in the Materials and Methods section, and point out the line numbers of the corresponding revisions in the revised manuscript.

Response 2.3R2: Thank you for pointing this out. The sentences „A total of 47 mice were used in the study: 24 miR-155 KO mice and 23 miR-155 sufficient mice (one mouse died during the experiment). All mice were female and between 8-12 weeks.” have been added to 4.1. Mice in the Materials and Methods section, p. 10-11, L 288-290.

(2) Comments 2.5: How many RNA samples were extracted from each group?
Response 2.5: Thank you for your question. For miRNA sequencing 6 RNA samples were isolated from 24 KO mice and 6 samples from 23 WT mice. In total, 12 RNA samples (three from each group: WT, WT_MOG, KO, and KO_MOG) were isolated. Each RNA samples was isolated from lymphocytes derived from three or four mice respectively.

The authors should describe this Response in the Materials and Methods section, and point out the line numbers of the corresponding revisions in the revised manuscript.

Response 2.5R2: Thank you for pointing this out. The fragment „For miRNA sequencing six RNA samples were isolated from 24 KO mice, and an-other six samples from 23 WT mice. Therefore, a total of 12 RNA samples (three from each group: WT, WT_MOG, KO, and KO_MOG) were isolated. Each RNA sample was isolated from lymphocytes derived from three or four mice respectively.” has been added and the sentence “Twelve RNA samples (three from each WT, WT_MOG, KO, and KO_MOG groups) were thoroughly evaluated according to Novogene’s quality standards.” has been changed to “RNA samples were thoroughly evaluated according to Novogene’s quality standards.” 4.7.1. Quality control in the Materials and Methods section, p. 13, L 398-402.

(3) Comments 2.7: What was “the genome website”?
Response 2.7: Thank you for this question. Reference genome (ensembl_mus_musculus_grcm38_p6_gca_000001635_8) and gene model annotation files were directly downloaded from Ensemble.

The authors should describe this Response in the Materials and Methods section, and point out the line numbers of the corresponding revisions in the revised manuscript.

Response 2.7R2: Thank you for pointing this out. The sentence was already in the manuscript, so it was justr clarified. Fragment " …..from the genome website” has been changed to „ …. from the Ensemble.” to 4.6.2. Bioinformatics analysis in the Materials and Methods section, p. 12, L 384.

(4) Comments 2.9: In the “4.6. mRNA sequencing”, “Eight RNA samples were rigorously assessed”. In the “4.7. miRNA sequencing”, “Twelve RNA samples (three each from WT, WT_MOG, KO, and KO_MOG groups) were thoroughly evaluated”. Please explain the difference between the sample numbers.
Response 2.9: Thank you for this question. According to the project plan we were focused on finding changes in miRNA profile between KO and WT samples after MOG stimulation (12 samples three from each: WT, WT_MOG, KO, and KO_MOG groups). mRNA sequencing was additional step (out of project plan). Only in 8 samples (two from each: WT, WT_MOG, KO, and KO_MOG) there was enough RNA for mRNA sequencing. Nevertheless, it provided a valuable feedback. That is why there is a discrepancy between sample number in both sequencing.

The authors should describe this Response in the Materials and Methods section, and point out the line numbers of the corresponding revisions in the revised manuscript.

Response 2.9R2: Thank you for pointing this out. The fragment „….(two from each: WT, WT_MOG, KO, and KO_MOG)….” has been added to 4.6. mRNA sequencing in the Materials and Methods section, p. 12, L 360-361.

(5) Comments 3.11: The Results of the mRNA sequencing were very preliminary. Please provide the detailed information of how many genes were significantly up-regulated or down-regulated in the CD4+ T cells after the MOG restimulation.
Response 3.11: Thank you for this question. Data from mRNA and miRNA sequencing of changes between KO MOG and WT was attached in supplementary materials. Differential expression analysis between conditions or groups with p adj. < 0,05. In CD4+ T cells after MOG restimulation from miR-155 deficient mice to control WT we observed upregulation of 23 genes and 16 genes were down-regulated.

The authors should describe this Response in the Results section, and point out the line numbers of the corresponding revisions in the revised manuscript.

Response 3.11R2: Thank you for pointing this out. The sentences „Data from mRNA and miRNA sequencing of changes between KO MOG and WT was attached in supplementary materials. Differential expression analysis between conditions or groups with p adjusted < 0,05. In CD4+ T cells after MOG restimulation from miR-155 deficient mice to control WT we observed upregulation of 23 genes and 16 genes were downregulated.” have been added to 2.5. Ffar1 and Scg2 upregulation in miR-155 deficient CD4+ T cells in the Results section, p. 7, L 164-168.

(6) Comments 3.12: Regarding the “To confirm the successful deletion of the targeted genomic sequence, PCR was performed”, did the authors sequence the amplified PCR products?
Response 3.12: Thank you for pointing this out. It is a good idea to sequence the PCR product, which would ascertain us with the successful deletion of miR-155 sequence. However, we did not sequence PCR product, we compared the sizes of the product on electrophoresis agarose gel. We will introduce this step in the future projects.

The authors should discuss this Response in the Discussion section, and point out the line numbers of the corresponding revisions in the revised manuscript.

Response 3.12R2: Thank you for suggestion. The sentence „PCR product was not sequenced, but the size of the product was confirmed on electrophoresis agarose gel; the results confirmed the deletion of miR-155, thus allowing the identification of…….” has been added to 2.2. Mutagenesis and miR-155 KO mice generation in the Results section, p. 3, L 96-98.
In addition, one of the paragraphs has been amended. The sentence: “In the present study, miR-155 was selected KO mice as the research model as miR-155 is believed  to be a critical regulator in the response of peripheral CD4+ T cells to myelin antigen stimulation.” has been deleted (with track p. 9, L 218-220) and a following section has been added: “In the present study, a novel miR-155 KO mouse as a research model to better understand the role of this miRNA as a critical regulator in the response of peripheral CD4+ T cells to myelin antigen stimulation. Successful deletion of miR-155 was confirmed by both modification of the genomic locus and by demonstration of the absence of miR-155 transcripts. Moreover, sequencing of the PCR product would further confirm successful deletion of the selected region.”, p. 9, L 231-236.

(7) Comments 3.14: Regarding the “two genes, Ffar1 and Scg2, showed increased expression”, what were fold changes of these genes?
Response 3.14: Thank you for your question. The expression of Ffar1 and Scg2 genes in miR-155-deficient CD4+ T cells stimulated with MOG was increased by approximately 6-fold and 9-fold, respectively, compared to WT cells.

The authors should describe this Response in the Results section, and point out the line numbers of the corresponding revisions in the revised manuscript.

Response 3.14R2: Thank you for suggestion. The following sentence „The expression of the Ffar1 and Scg2 genes were increased by approximately 6-fold and 9-fold, respectively, in miR-155-deficient CD4+ T cells stimulated with MOG compared to WT cells.” has been added to 2.5. Ffar1 and Scg2 upregulation in miR-155 deficient CD4+ T cells in the Results section, p. 7, L 172-174.